# Virtual HDR Boost for Prostate Cancer: Rebooting a Classic Treatment Using Modern Tech

**DOI:** 10.3390/cancers15072018

**Published:** 2023-03-28

**Authors:** Eric Wegener, Justin Samuels, Mark Sidhom, Yuvnik Trada, Swetha Sridharan, Samuel Dickson, Nicholas McLeod, Jarad M. Martin

**Affiliations:** 1School of Medicine and Public Health, The University of Newcastle, Callaghan, NSW 2308, Australia; 2Department of Radiation Oncology, Calvary Mater Newcastle Hospital, Waratah, NSW 2298, Australia; 3GenesisCare, Maitland, NSW 2323, Australia; 4GenesisCare, Gateshead, NSW 2290, Australia; 5Department of Radiation Oncology, Liverpool Hospital, Liverpool, NSW 2170, Australia; 6Department of Urology, John Hunter Hospital, Newcastle, NSW 2305, Australia

**Keywords:** prostate cancer, stereotactic body radiotherapy (SBRT), review, boost, brachytherapy

## Abstract

**Simple Summary:**

Brachytherapy for prostate cancer is a method where radiotherapy is directly delivered to the prostate via surgical insertion of the radioactive source. Brachytherapy can increase the amount of radiation delivered to the prostate cancer but sometimes at the cost of increased side effects. Here, we review promising early results from alternative non-invasive techniques that now exist and can deliver similar radiotherapy doses without the need for the surgical procedure required for brachytherapy.

**Abstract:**

Prostate cancer (PC) is the most common malignancy in men. Internal radiotherapy (brachytherapy) has been used to treat PC successfully for over a century. In particular, there is level-one evidence of the benefits of using brachytherapy to escalate the dose of radiotherapy compared with standard external beam radiotherapy approaches. However, the use of PC brachytherapy is declining, despite strong evidence for its improved cancer outcomes. A method using external beam radiotherapy known as virtual high-dose-rate brachytherapy boost (vHDRB) aims to noninvasively mimic a brachytherapy boost radiation dose plan. In this review, we consider the evidence supporting brachytherapy boosts for PC and the continuing evolution of vHDRB approaches, culminating in the current generation of clinical trials, which will help define the role of this emerging modality.

## 1. Introduction

Radiotherapy is an effective treatment option for men with prostate cancer (PC), with disease control outcomes similar to what can be achieved with surgery [1]. Radiotherapy can be delivered either externally (external beam radiotherapy (EBRT)) or internally (brachytherapy). The advantage of brachytherapy is in the ability to deliver high radiation doses with rapid dose falloff, which improves the potential for tumour control while minimizing the risk of damage to adjacent healthy tissues. Increased radiotherapy doses have been found to be associated with higher tumour control probability (Figure 1). This relationship has been demonstrated in several randomized trials, which have reported better disease control outcomes with higher radiotherapy doses [2]. 

Brachytherapy monotherapy is a proven treatment modality for favourable intermediate risk PC based on NCCN treatment guidelines, with excellent oncological outcomes [3,4]. Higher risk PC necessitates a combined modality approach with brachytherapy boosting and EBRT due to the risk of disease outside the prostatic capsule. 

Brachytherapy has been used as a radiation dose-escalation strategy in several randomized trials, consistently demonstrating improved disease control compared with standard doses of EBRT [5,6,7]. Despite this, the use of brachytherapy in the management of PC has markedly declined internationally [8,9]. Simultaneously, technological advances have led to advanced EBRT techniques emerging, which have the ability to deliver higher doses of radiation accurately over a small number of treatment sessions or fractions. This technique is called stereotactic body radiotherapy (SBRT), and it has been widely investigated for the management of PC over the last two decades [10,11,12]. This has naturally led to efforts to apply SBRT techniques in an attempt to mimic the outcomes that can be achieved with brachytherapy. In this review, we focus on the biology and clinical evidence behind dose escalation for PC, aiming at a target audience consisting of all clinicians involved in the management of this disease. Furthermore, we explore the evidence supporting the employment—and, conversely, the declining use—of brachytherapy to achieve this, as well as the investigations into SBRT as an alternative to brachytherapy, which is termed virtual high-dose-rate brachytherapy boost (vHDRB). Our search strategy pursued to achieve this is outlined in Figure 2.

### Selection Criteria

Titles and abstracts of records were searched to assess inclusion. The radiotherapy studies needed to combine a conventional schedule with a ”boost” schedule, with a high dose per fraction size. These records needed to report a cancer control endpoint and/or late toxicity endpoint. The records also needed to report the radiation dose used. If the abstract and/or title had details suggestive of the inclusion criteria, the full study was obtained and reviewed for further analysis. The criteria excluded records involving stereotactic monotherapy, simultaneous integrated boost (SIB), or radiotherapy delivered in the salvage setting. Only boost studies were selected for review, as this radiotherapy technique has emerged more recently. Both these techniques developed alongside one another; however, we await the results of randomized trials to adequately compare these two [13]. Studies that lacked significant details were also excluded from the analysis. For instance, in a multi-institutional patient registry study [14] that involved 437 patients, only a small proportion (5%) of the patients received a stereotactic boost; therefore, no conclusive findings could be drawn regarding this subgroup. 

Thirty-four studies were found to meet the search criteria. Some study populations were described in more than one publication, such as in the work by Kim et al. [15]. For these 34 relevant studies, each publication’s references and citations were further searched and reviewed to assess the strength of the original search strategy. Google Scholar and Scopus were used for the citation search, resulting in a total of 680 citations. No further eligible records were discovered using this citation/reference search. 

## 2. Radiation Dose Escalation—Theory and Evidence

Radiotherapy has been used for PC treatment for over a century. The rationale for dose escalation is that it improves tumour control probability (TCP). Conversely, increased radiation doses to healthy structures lead to higher rates of toxicity, a relationship described by the normal tissue complication probability (NTCP). There is, therefore, always a trade-off between maximizing TCP and ensuring NTCP is not unacceptably high. 

Radiotherapy originally evolved from treatment close to normal tissue tolerance. With new technological advancements, higher radiotherapy doses could be given safely and/or NTCP decreased. A significant step forward in the development of EBRT was the advent of three-dimensional conformal radiation therapy and intensity-modulated radiation therapy (IMRT), where the radiotherapy beams are more precisely shaped to the desired target volume. With these developments, dose escalation trials began as early as the 1980s [16]. EBRT dose escalation evolved from a 2 Gy per fraction approach, with improved biochemical control seen across low-, intermediate-, and high-risk PC. The increasing dose has led to better biochemical control, with a plateau yet to be seen [17,18]. An MD Anderson Cancer Centre randomized phase 3 trial from 1993 to 1998 compared a 70 Gy arm to a 78 Gy arm. Using a relatively simple four-field box technique resulted in improved 15 year clinical control (hazard ratio (HR): 0.61, *p* = 0.042), distant metastasis-free survival rates (HR: 0.33, *p* = 0.018), and prostate cancer-specific survival (HR: 0.52, *p* = 0.045) in the 78 Gy arm [19]. The RTOG 0126 randomized trial compared a dose escalation of 70.2 Gy to 79.2 Gy, and 33.8% of the patients were managed with the more contemporary IMRT approach, where the treatment beams are dynamically shaped to better conform to the shape of the desired target. This trial showed, for intermediate-risk PC, improved 8 year biochemical control (HR: 0.54, *p* < 0.001), as well as a reduced need for salvage therapy (HR 0.63, *p* < 0.001), in the dose escalation arm [20]. These trials consistently showed increased rates of rectal and/or urinary toxicity, leading to efforts such as rectal spacing and image-guided treatment techniques being introduced to successfully offset such risks [21,22].

However, in radiotherapy, not every gray (Gy) may be equal, and new radiotherapy methods have been trailed to deliver dose escalation through different approaches. To comprehend this development, it is helpful to have an understanding of the biologically effective dose (BED). In short, a higher dose per fraction size can deliver a higher BED compared to conventional EBRT dose schedules of 2 Gy per fraction. Each tissue and tumour has its own radiation fraction size sensitivity described by the alpha/beta ratio. By using a higher dose per fraction size, theoretically, this should result in a higher biological dose being delivered to the tumour and lead to greater tumour cell death. Table 1 shows examples of BED dosing for different delivered fraction sizes. 

## 3. Brachytherapy as a Dose-Escalation Strategy

Around a century ago, brachytherapy using radium, which involved permanent low-dose-rate (LDR) implants, emerged as the first successful radiotherapy modality for prostate cancer treatment. Despite the changes in isotopes used, LDR monotherapy has consistently demonstrated excellent oncological outcomes in patients with favourable intermediate-risk PC. In the 1980s, as external beam radiation therapy (EBRT) techniques, such as 3D conformal radiotherapy, evolved, brachytherapy also saw technological advancements, including the introduction of temporary high-dose-rate (HDR) insertions. Following these developments, the combination of these two techniques was pursued to improve oncological outcomes in patients with higher risk PC. The approach involves using brachytherapy to intensify doses in areas with higher disease burden while delivering lower doses via EBRT to microscopic areas. This approach has shown great promise in improving oncological outcomes in patients with higher risk PC. There are three randomized control trials comparing EBRT to EBRT with brachytherapy boost, which have reported improved biochemical control in the brachytherapy boost arms [5,6,7].

The ASCENDE-RT randomized trial compared dose-escalated EBRT (78 Gy in 39 fractions) to EBRT (46 Gy in 23 fractions) + an LDR brachytherapy boost of 115 Gy using Iodine-125 implants. In the brachytherapy boost arm, the 10 year risk for biochemical failure was reduced compared to the dose-escalated EBRT arm (85% vs. 67%, *p* < 0.001) [24]. The ASCEND-RT trial did report increased toxicity rates in the brachytherapy boost arm compared to the dose-escalated EBRT arm [25]. At 5 years, there was a significant difference in the cumulative incidence of severe grade-three genitourinary (GU) toxicity: 18.4% for the LDR boost arm versus 5.2% in the dose-escalated EBRT arm. These included outcomes such as incontinence and need for catheterization. This trial also summarized toxicities from other brachytherapy boost trials (both HDR and LDR), which showed a wide range in severe late GU toxicity incidence from 1.4% to 31% [25]. 

## 4. Diminishing Role of Brachytherapy

Despite the excellent advantages in efficacy from using brachytherapy boost as a dose-escalation strategy, the practice has not been universally adopted and, indeed, is showing a marked decline [8,9]. MD Anderson, in a study from 2010 to 2015, reported that use of prostate brachytherapy was significantly declining both as a monotherapy and as EBRT combination therapy [26]. EBRT use increased during this timeframe. This was not unique and correlated to declining use across Australia [27]. Older age (>70), higher risk subgroups, and treatment at an academic centre correlated with decreased brachytherapy utilization. 

The reason for the declining brachytherapy use is likely multifactorial. One factor is the concern amongst radiation oncologists about the high toxicity rates that brachytherapy may cause, resulting in outcomes such as urethral stricture formation. The technical expertise required for optimal outcomes can be a challenge to obtain, with improper needle insertion causing urethral traumatization and catheter migration contributing to higher organ-at-risk (OAR) doses. Further factors may include poor financial remuneration, decreased trainee exposure, and travel distance to expert centres. Further challenges faced by brachytherapy uptake include poor concordance amongst groups in terms of isotope use, prescribed dose, and integration with androgen deprivation therapy (ADT) and a relative lack of level-one evidence comparing EBRT with brachytherapy boost to dose-escalated EBRT. As brachytherapy use has declined, improved EBRT techniques have come under development, such as SBRT, which potentially allows for dose escalation using a less invasive approach. 

## 5. Emergence of SBRT

Early monotherapy SBRT trials looked at keeping an isoequivalent BED using a pure SBRT approach. The HYPO-RT-PC phase 3 randomized controlled trial (RCT) [11] compared 78 Gy in 39 fractions with 42.7 Gy delivered over seven sessions and showed equivalent 5 year biochemical PFS in both approaches at 84% (95% CI: 80–87), with an adjusted HR of 1.002 (95% CI: 0.758–1.325; log-rank *p* = 0·99). The more common five-fraction approach is also being investigated, with early results available from the PACE-B phase 3 RCT [12], which compared 78 Gy in 39 fractions with 36.25 Gy in 5 fractions. This trial showed that the 2 year RTOG toxicity rates were similar between the two arms, with efficacy data expected in 2023. 

Virtual HDR radiotherapy monotherapy trials started as early as 2008, with Fuller [28] undertaking a ten-patient pilot trial utilizing a CyberKnife platform. Shortly after, several small vHDRB trials started between 2010 and 2012, with early investigators including Mirallbel (Spain), Oermann (USA/Washington), Anwar (USA/San Francisco), and Katz and Kang (USA/New York). 

This literature review search retrieved 34 relevant records for review. Some researchers used the same study population to publish multiple times. When this was accounted for and these studies were conglomerated, 26 unique patient populations were left. The characteristics and results of these studies are summarized in Table 2. 

### 5.1. Biochemical Progression-Free Survival

Despite the large heterogeneity amongst the trials listed in Table 2, excellent bcPFS and other closely associated endpoints can be noted. vHDRB studies included low- to very-high-risk PC patients, with efficacious biochemical control seen across the groups. Paydar [41] showed a 3 year bcPFS of 100% for intermediate-risk PC. Kim’s study [15] without ADT showed an 8 year bcFFS of 100% for intermediate-risk PC and 77.8% for high-risk PC. However, this needs to be taken in the context of the small numbers in this study, which had only 11 high- and 31 intermediate-risk patients. Larger, prospective phase 2 multicentre trials are listed in Table 2, such as the CKNO-PRO and the PROMETHEUS trials. For example, the latter included a mixture of 135 intermediate- and high-risk patients with 2 year bcPFS of 98.6%. 

These results are similar to conventional radiotherapy, with high-risk PC patients in the HYPO-PROST trial randomized to a vHDRB arm and conventional dose arm and reports of 5 year bcRFS being 78.2% and 82.9% [57]. This closely follows the ASCENDE-RT brachytherapy boost arm in terms of outcomes.

There are no strong randomized data to compare HDR brachytherapy boost to vHDRB, and future randomized control trials are needed to answer this. The closest data to compare these two approaches come from a propensity score-matched analysis by Chen [59]. This study included 131 patients for the vHDRB and 101 patients for the HDR brachytherapy boost. The median follow-up was 73.4 months for vHDRB and 186 months for the HDR brachytherapy boost. One of the study’s strengths lay in accounting for a large number of known covariates. The majority of PC patients included were high- and very-high-risk PC patients, as defined by NCCN criteria. The five- and ten-year unadjusted bcRFS rates were 88.8% and 85.3% for the vHDRB compared to 91.8% and 74.6% for the HDR boost brachytherapy. Metastasis-free survival was also analysed and, again, no statistical significance was seen in the difference between the two groups. The vHDRB used two dose schedules, 19 Gy and 21 Gy, both in two fractions, with no statistically significant differences between these two. 

Two studies appeared to be divergent from the rest in terms of poor disease control. Koh’s study [38], reported a 50% bcPFS at a median cohort follow-up of 29.3 months. This may have been due to the inclusion of metastatic disease patients and low patient numbers. Khmelevsky [35] reported a 60% 5 year bcRFS, which may have related to the initial proton use and different boost schedules. 

Most of these studies staged patients with conventional CT and bone scans. Radiomics has the potential to offset some of the poor sensitivity and specificity issues [62]. PSMA PET staging has been shown to be superior compared to conventional staging [63], and future trials may be expected to have better biochemical control due to improved patient selection. 

Despite the heterogeneity in these trials in the magnitude of factors, the oncological results are concordant and excellent. These trials spanned multiple countries and institutions. Certain questions and challenges remain in defining outcomes and measurable endpoints: first, the varied dose schedules and dose distributions, and second, cofounders, such as ADT use/duration and pelvic nodal radiotherapy inclusion. Different endpoints were used in these trials and the definition for biochemical relapse varied, although Phoenix criteria were most commonly used. Many studies were small in patient numbers and had short follow-up times. 

### 5.2. Toxicities

The use of a high dose per fraction size has a long history in radiotherapy, with increased toxicity rates [64]. Table 2 illustrates that, amongst the vHDRB trials, toxicities were generally much lower. 

Acute grade 2 (G2) GU toxicities with vHDRB ranged from 17% to 47%, with grade 3 (G3) ranging from 0% to 4% [65]. Acute GI toxicity with vHDRB had a range of 0% to 21% for G2, with no acute G3 being identified [65]. These toxicity rates are similar to that of dose-escalated EBRT. 

These studies reported late GU G2 toxicity ranging from 1% to 25% and G3 from 0% to 5%. Most patients in these studies had a late G3 GU toxicity of less than 3%. Late G2 GI toxicities ranged from 0% to 18% and G3 from 0% to 4%. 

vHDRB late toxicity data differed from those for conventional fractionation radiotherapy. There appeared to be a flare in subacute toxicity around the 12–18 months range [52]. This was similar to the brachytherapy toxicity data from the ASCENDE-RT study [25]. Even though the cumulative incidence was high in this trial over time, most of the subacute toxicity resolved by the 2 year mark. Interestingly, this 12 month GU flare toxicity seemed to be lower in vHDRB compared to SBRT monotherapy (7.60 ± 0.42 and 9.53 ± 0.47, *p* = 0.003), as reported by Feng using the American Urological Association (AUA) symptom index [48]. Katz and Kang [31] may also support this, with lower late GU G3 toxicity seen in the vHDRB group compared to the SBRT monotherapy arm (2.3% versus 3.9%) and 2.3% versus 7.8% for late GU G2 toxicity. These data were nonrandomized, and contradictory observations have also been cited from other institutions [37]. Chen [59], using a propensity score-matched analysis to compare vHDRB to HDR brachytherapy boost, found no significant difference in the combined rates of G3+ toxicities between these two groups. 

There were five patients in total identified as exhibiting grade-four toxicity. Novikov [61] reported 3 patients out of 51 who developed severe rectal toxicity and required diversion colostomies. One of these three patients also had significant bleeding requiring ICU support. Moreover, one patient in Alayed’s study [49] experienced a rectal fistula, which repeated rectal biopsies were thought to have contributed to. The last patient in Pollack’s study [53] developed sepsis after post-treatment transurethral resection. 

Toxicity data from all these studies have multiple contributing factors that are not easily teased out. An example is pelvic nodal irradiation and the effect on GI toxicity. Some studies combine G2+ toxicities, whereas others divide G2 and G3. There were differences in scoring and statistical calculation methods. Some series reported toxicity as the prevalence at a specific time-point while others cited a cumulative toxicity result. Overall, these data support the safety profile for using vHDRB for PC, and this treatment modality may have lower late toxicities compared to brachytherapy. The main caveat is that much of the data came from relatively small, single-institution series and were nonrandomized.

## 6. SBRT as a Dose-Escalation Strategy—Virtual Boosting

vHDRB can be delivered through a variety of methods. The first key variable is the platform. Although there are various subcategories within each of the following classes, platforms can be broadly categorized as CyberKnife (CK), linear accelerator, or protons/other heavy particles. CK is a unique, compact linear accelerator design with the treatment head mounted on a manoeuvrable robotic arm that allows for radiotherapy delivery to be undertaken at a large number of non-coplanar angles. Each radiotherapy beam delivers a small cylindrical radiotherapy dose profile, with radiation plans being composed of over 100 of these small columnar beams. CK has a unique stereoscopic image guidance system that directs the beam to improve delivery accuracy with motion. 

The most common of all radiotherapy machines is the standard linear accelerator. With modern linear accelerators, dose delivery tends to be undertaken with volumetric modulated arc therapy. Radiotherapy is delivered using two to three beam planar arcs, with dose modulation employed from multileaf collimators. Lastly, there are heavy particles, with protons being the most commonly used. Heavy particles theoretically have an improved radiobiology profile compared to the photons described above. Compared to photons, protons have the ability to deliver a large dose at a certain depth. This ability allows for improved sparing of normal tissue; however, the ability to shape the dose tends to be more limited compared to photons. 

The second variable is the dose/radiotherapy schedule, and a wide variety of schedules can be seen, as illustrated in Table 2, with the common theme of trying to mimic the physical brachytherapy boost approaches given over two to four sessions. 

The last variable is the dose distribution, which varies between centres and trials. Some methods employ a relatively homogeneous whole-gland dose distribution (Figure 3A), whereas others aim to introduce inhomogeneity to replicate a brachytherapy implant with the purposeful aim of creating “hotspots” within defined volumes to receive up to 150% of the prescribed dose. These virtual volumes are intended to direct hotspots towards brachytherapy-type structures, such as spheres or cylinders. The CK platform can better replicate a virtual cylinder dose distribution compared to a standard LINAC due to the large number of non-coplanar angles used during delivery. A “hot shell” distribution aims to create hotspots along the prostate periphery with central urethra cooling (Figure 3B). 

Another method gaining traction due to the FLAME RCT [66] is dominant intraprostatic lesion (DIL) boosting, where the boost dose is concentrated in the image-defined cancer volume (Figure 3C). FLAME used a simultaneous integrated boost and delivered a dose of up to 95 Gy in 35 fractions to the MRI DIL, with the whole prostate gland receiving 77 Gy in 35 fractions. A total of 84% of these patients were high risk and two thirds received ADT for up to 3 years. This trial reported improved 5 year biochemical disease-free survival (bcDFS), increasing from 85% to 92%, in the boost arm (HR: 0.45, 95% CI: 0.28 to 0.71, *p* < 0.001) without any significant increase in late GI or GU toxicities. The BOOSTER trial [50] integrated DIL boost with vHDRB, with three dose levels given. After the conventional radiotherapy dose of 46 Gy in 23 fractions, the boost used 20 Gy, 22 Gy, or 24 Gy to the prostate and, correspondingly, 25 Gy, 27.5 Gy, and 30 Gy to the DIL, all in two fractions. The highest dose of 30 Gy in two fractions was modelled to reflect the 150% isodose distribution of an HDR plan but preferentially directed towards the DIL. 

## 7. Technical Issues with Virtual Boosting

The ability to implement such high radiotherapy doses per fraction size is linked with the development and application of technological advancements. These can be subdivided into three categories; improved target delineation, enhanced accuracy of treatment delivery, and the use of organ-at-risk (OAR) stabilizers or physical barriers. The stereotactic boost trials shared some similarities with these technological advancements. They often used MRI to better delineate structures, fiducials to help improve accuracy through inter- and intra-fraction motion management, and bladder-filling/rectal-emptying/low-gas-diet protocols.

### 7.1. Simulation Imaging

Due to its improved ability to accurately define the prostate and other relevant parts of the anatomy [67], MRI has helped substantially in safely delivering stereotactic radiotherapy. The PROMETHEUS trial with 135 patients employed MRI fusion for radiotherapy planning. MRI helps with delineation of all structures, including the DIL (Figure 4). This was a common feature in all of these vHDRB studies. Future studies can also be expected to utilize PSMA staging for both more accurate staging and DIL delineation. 

### 7.2. Image Guidance

Intrafraction image guidance is another technological advancement that has helped in the delivery of stereotactic radiotherapy doses whilst minimizing GI and GU toxicity. With fiducial placement and IGRT planning, treatment volumes can be reduced substantially. Most of the trials in Table 2 required fiducial placements and image guidance.

A variety of systems can be employed to help safely reduce margins and improve radiotherapy targeting accuracy. CyberKnife was one of first of such systems utilized. Calypso uses inserted radiofrequency transponders to track and adjust radiation treatment beams. The Varian Truebeam gating is another integrated package commonly used for prostate intrafraction motion management to help enable precision fiducial targeting. Other approaches are also being assessed, such as kilovoltage intrafraction monitoring (KIM), as validated in the TROG 15.01 SPARK trial [10,68]. This technology allows for image verification X-rays to be obtained in real time while prostate radiotherapy is being delivered. One of the advantages of KIM is that it enables strategies such as patient shifting or beam shifting during treatment. KIM has already been shown to have improved treatment accuracy, with the average systematic accuracy measured at 0.46 mm [69], and has the potential for further reductions to the treatment planning margins. As a “proof of principle”, the latter three systems described above were all successfully utilized in the BOOSTER vHDRB trial [50].

The trials mentioned in Table 2 tended to have PTV margins from 5 to 7 mm isometrically, except for the 3 to 5 mm posterior. The exception was the Jabbari/Anwar trial, which utilized a 0 mm PTV margin for part of the cohort and 2 mm isometric margin for the rest. These tight margins are only potentially achievable using real-time image-guided radiotherapy, with recent data suggesting that an MRI-LINAC may permit uniform margin reductions to 2 mm for prostate SBRT, although longer term efficacy data are required to confirm there is no loss of efficacy with this aggressive margin reduction [70].

### 7.3. OAR Stabilization Devices and Hydrogel Spacers

Endorectal devices were utilized in some of these trials to help minimize rectal toxicity. The endorectal balloon works by inflating a rectal balloon near the prostate, which decreases the radiotherapy dose to the rectum and helps avoid circumferential radiotherapy dose delivery. Miralbell [29] used this approach, stating concerns that the higher late GI toxicities seen may have been caused by the balloon pushing the anterior rectal wall closer to the high-dose prostate radiotherapy region. It was also seen that, in one third of patients, the rectal balloon exhibited systematic errors due to air leakage, suboptimal inflation, or suboptimal rectal emptying.

A rectal displacement device is a rigid device that aims to reduce variations in rectal filling and intrafraction movement. It works by pulling the posterior rectal wall away from the prostate and thereby decreasing the radiation dose to the rectum (Figure 5A) [14,71]. 

Hydrogel spacer placement is an alternative method employed to minimize rectal toxicity. The anterior rectal wall lies in close proximity to the prostate, with only 2 to 3 mm typically separating the target volume from this dose-limiting OAR [72]. Injection of a hydrogel spacer to increase this distance can be employed (Figure 5B). This allows the higher radiation doses to fall in the spacer region instead of the anterior rectal wall. A systematic review and meta-analysis on perirectal hydrogel spacer placement with 1011 patients [73] showed that this method was associated with less rectal irradiation, fewer GI toxicities, and higher bowel-related quality of life in the long-term follow-up. This study also showed with 486 patients that the space between the rectum and prostate had a median distance of 11.2 mm. In a multicentre trial, 222 patients were randomized to hydrogel injection and no injection [74]. The trial demonstrated a decrease in rectal dose and a significant reduction in late rectal toxicity (2.0% versus 7.0% in the control group, *p* = 0.04) with the use of hydrogel insertion.

### 7.4. Radiotherapy Platform

The trials identified in Table 2 used mixed radiotherapy delivery systems. Some studies used CyberKnife and others used conventional LINAC. Radiotherapy boost was utilized in the proton and carbon ion settings. The results showed no obvious indications as to which platform performed better with regard to the oncological outcomes. For this to be seen, one would assume that much larger patient numbers would be needed. The PACE-B trial [12] mentioned some differences in toxicities seen between CyberKnife and standard linear accelerator approaches; however, the low numbers precluded statistical analysis. The NINJA RCT included both CK and LINAC patients, so it should be informative as to whether any clinically relevant differences exist between these two particular approaches [13]. 

## 8. Future Directions

High-quality prospective randomized data are needed before vHDRB can be considered the standard of care. Currently active trials were identified using the ClinicalTrials.gov database with the disease term “Prostate cancer” and the other term “boost”. This resulted in 162 studies for screening. The same eligibility criteria as listed above were applied. Fourteen relevant trials were identified from the ClinicalTrials.gov database that involved delivery of a vHDRB for treatment of PC. These results are shown in Table 3. 

Four randomized trials stood out as helping to evaluate the true impact of a vHDRB on PC. The first was NCT03380806 from Hamilton, Canada. This trial of 100 high-risk patients looked into delivering conventional dosing of 45 Gy in 25 fractions, then randomized patients to either further conventional radiotherapy of 32–33 Gy in 15–16 fractions or a vHDRB of 19.5–21 Gy in 3 fractions. ADT will be given for a total of 3 years and outcome measures will be quality of life, safety, and efficacy. 

The second study (NCT01839994), from Gliwice, Poland, aimed to recruit 350 intermediate- and high-risk patients to a phase 3 randomized trial. The trial aimed to randomize patients to a cohort with 76–78 Gy in 38–39 fractions and a cohort with 50 Gy in 25 fractions + boost. The boost radiotherapy decision will be made using a nonrandomized approach between either HDR brachytherapy (20 Gy in two fractions) or vHDRB (20 Gy in two fractions). ADT will also be given for 3 years for high-risk patients. The primary endpoint will be freedom from biochemical failure at 3 years. The trial allows for PSMA PET staging. This study will hopefully shed more light on the difference between SBRT vs. brachytherapy boost. Another future trial that will help discern the differences between brachytherapy and SBRT is the ASCENDE-SBRT trial [75]. This trial aims to randomize 710 unfavourable intermediate-risk and high-risk PC patients to either a schedule with whole pelvic radiotherapy + brachytherapy boost with 46 Gy in 23 fractions or an ultra-hypofractionated schedule of 25 Gy in 5 fractions administered to the whole pelvis + a vHDRB boost of 40 Gy in 5 fractions. 

The final study (NINJA/TROG 18.01) is an international, multicentre trial looking to randomize 472 unfavourable intermediate-risk and high-risk patients between a monotherapy SBRT arm with 40 Gy in 5 fractions and an arm with 36 Gy in 12 fractions that includes a 20 Gy vHDRB in 2 fractions. For high-risk patients, PSMA PET staging is included. Another new technological advancement in this study is the use of only MRI planning. Radiotherapy plans are created with a synthetic CT derived from the MRI [76,77]. ADT is given for 6 months. The primary endpoint will be biochemical control at 5 years. This study will aim to tease out whether there are any efficacy and/or toxicity differences between SBRT monotherapy and vHDRB. 

## 9. Conclusions

This review summarizes results from the published literature regarding the use of vHDRB in the treatment of PC. Although most of the series are relatively small, single-centre trials, emerging multicentre data from larger series with longer follow-ups suggest that vHDRB has high efficacy and a favourable toxicity profile. Current randomized trials will help determine whether vHDRB has a wider role as an option in the future standard management of prostate cancer. 

## Figures and Tables

**Figure 1 cancers-15-02018-f001:**
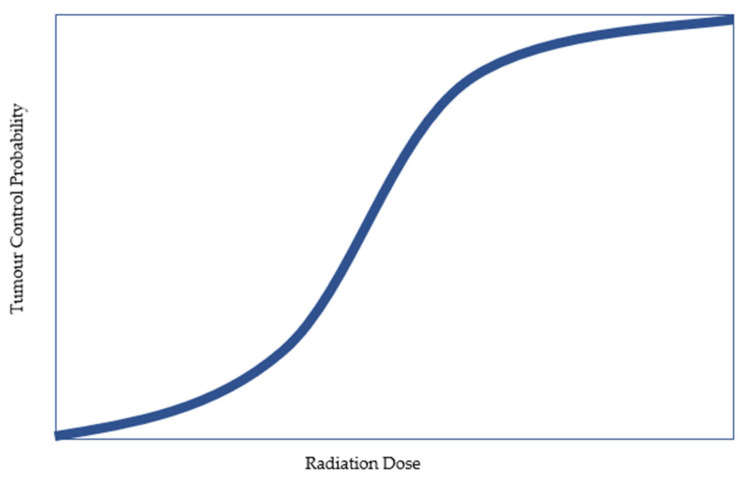
Idealized tumour control probability curve.

**Figure 2 cancers-15-02018-f002:**
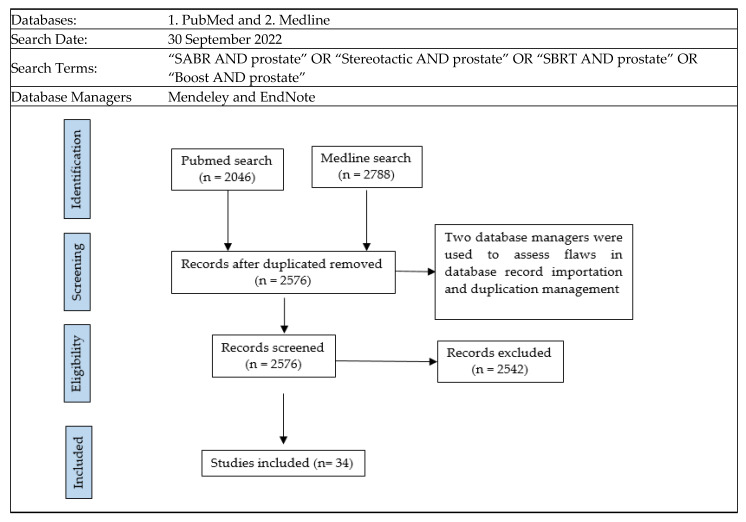
Search strategy.

**Figure 3 cancers-15-02018-f003:**
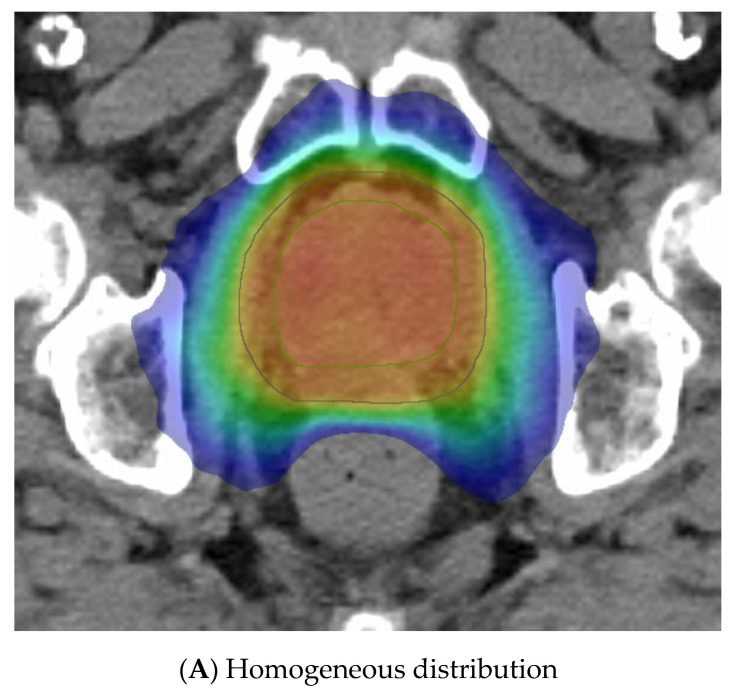
Examples of virtual high-dose-rate brachytherapy boost radiotherapy dose distributions, with the colour gradient demonstrating 50% of the dose in blue up to the maximum doses in red. The green outline represents the prostate, the blue outline the planning target volume (PTV), yellow is the urethra, and burgundy the DIL, when outlined. (**A**) Homogenous distribution with 100% of the dose in red. (**B**) Hot-shell distribution showing the red 150% dose directed towards the prostate peripheral zone, with lime being the 100% dose. (**C**) Dominant intraprostatic lesion distribution, with the 150% dose in red directed towards the DIL and 100% dose in lime covering the PTV.

**Figure 4 cancers-15-02018-f004:**
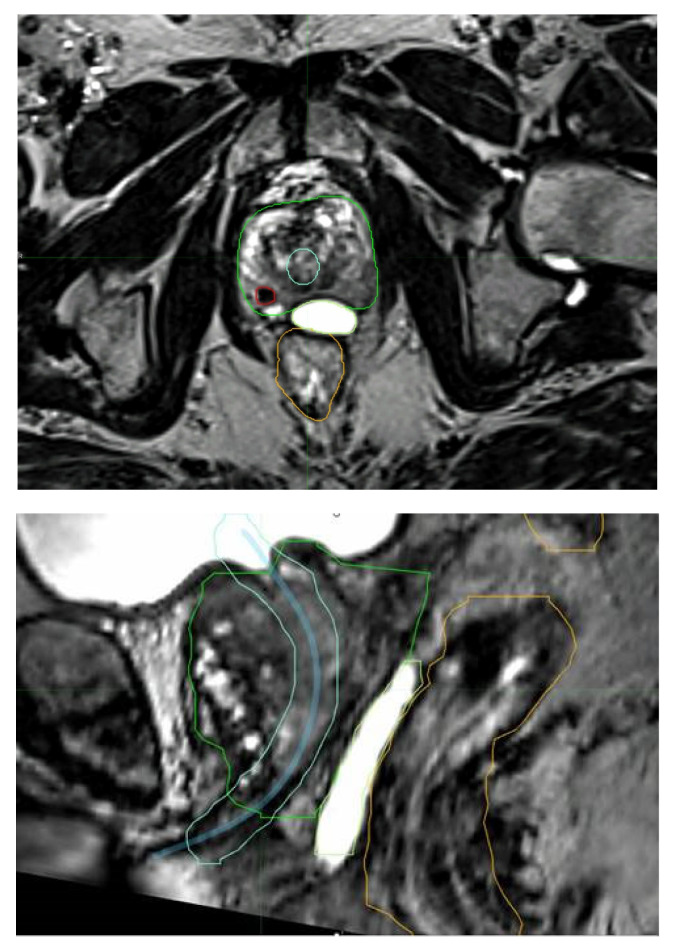
MRI fusion aiding with the identification of a dominant intraprostatic lesion (red), the urethra with expansion (blue and light blue), rectal sparing hydrogel (lime), the rectum (orange), and the prostrate (green).

**Figure 5 cancers-15-02018-f005:**
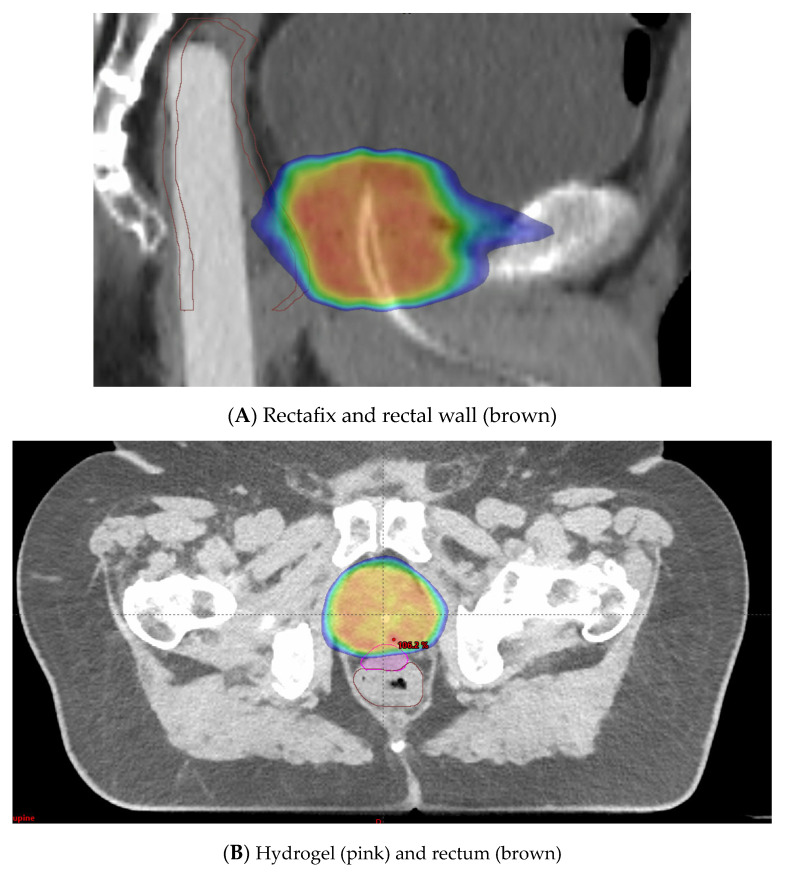
(**A**) Rectafix in use to mitigate circumferential rectal radiotherapy doses. (**B**) Hydrogel in use to prevent high radiotherapy doses to anterior rectal wall. Blue is 50% isodose distribution. Red is 100% isodose distribution.

**Table 1 cancers-15-02018-t001:** BED examples using a prostate cancer alpha/beta ratio of 1.5 [23].

Radiotherapy Schedule	Total Dose (Gy)	Dose/Fraction (Gy)	Number of Fractions	Prostate Cancer BED_2_ (Gy)
Conventional (non-dose escalated)	68	2	34	159
Moderate hypofractionation	60	3	20	180
Conventional (dose escalated)	78	2	39	182
Conventional + SBRT boost	46 19	2 9.5	23 2	107 139 Total: 246

**Table 2 cancers-15-02018-t002:** Studies using vHDRB.

**Author/Study**	**Year**	**N**	**Risk**	**Median Follow-Up (Months)**	**Conventional Dose**	**Pelvic Nodal RT**	**ADT use and Duration**	**Boost Dose** **(Target)**	**Platform** **(CK, LINAC, Other Particle)**	**Endpoint**	**Late Toxicity**	**Additional Information**
Miralbel [29]	2010	50	5—LR 12—IR 33—HR D’Amico	72	64 Gy/32#	Some 56%	66% 6 months	10–16 Gy/2# (DIL)	LINAC	98% bcDFS	≥Gr 3 GU 0% GI 10% ≥Gr 2 GU 12% GI 20% RTOG	Pilot study
Oermann [30]	2010	24	13—IR 11—HR NCCN	9.3 (month average)	50.4 Gy/28#	Unknown	42% Unknown duration	19.5 Gy/3# (prostate and SVs)	CK	Not reported	No late GI or GU toxicities reported (limited follow-up) CTCAE	Pilot study
Katz and Kang [31,32]	2010 and 2014	45 in vHDRB cohort and 52 in SBRT monotherapy cohort	HR NCCN	vHDRB cohort 69	vHDRB cohort 45 Gy/25#	vHDRB cohort Yes 100%—partial coverage using four-field box	vHDRB cohort 62% Unknown duration	18–21 Gy/3# (prostate and SVs)	CK	vHDRB cohort 89.5%—IR 77.7%—HR (3 year bFFS) 69% (6 year bcDFS)	vHDRB cohort ≥Gr 3 GU 2.3% GI 0% Gr 2 GU 2.3% GI 13.3% RTOG	Retrospective, 2010 data included larger numbers due to inclusion of intermediate-risk PC
				SBRT monotherapy cohort 48	SBRT monotherapy cohort 35–36.25 Gy/5#	SBRT monotherapy cohort No	SBRT monotherapy cohort 50% Unknown duration			SBRT monotherapy cohort 6 year bcDFS—no difference between cohorts (*p* = 0.86)	SBRT monotherapy cohort ≥Gr 3 GU 3.9% GI 0% Gr 2 GU 7.8% GI 0% RTOG	
Jabbari [33] and Anwar [34]	2012 and 2016	48	14—IR 34—HR NCCN	42.7	45–50 Gy/25#	Some (if risk >15%, Roach formula)	88% 6 months	19–21 Gy/2# (prostate and SVs)	CK	90% (5 year bNED)	≥Gr 3 GU 2.1% GI 0% ≥Gr 2 GU 25% GI 0% CTCAE	Pilot study
Khmelevsky [35,36]	2012 and 2018	116 in vHDRB cohort and 173 in conventional cohort	LR—HR NCCN	vHDRB cohort 67.8	vHDRB cohort 44–46 Gy/22–23#	Some (selected IR—HR)	vHDRB cohort 95% 6 months	28.0–28.8 Gy (radiobiological equivalent of gray)/3–8#	Protons	vHDRB cohort 60% (5 year bcRFS)	vHDRB cohort ≥Gr 3 GU 2.8% GI 0.9% Gr 2 GU 8.3% GI 10.2% RTOG	Randomized
				Conventional cohort 71.6	Conventional cohort 68–72 Gy/34–36#		Conventional cohort 95% 6 months			Conventional cohort 61.9% (5 year bcRFS)	Conventional cohort ≥Gr 3 GU 3.8% GI 1.3% Gr 2 GU 9.1% GI 34.8% RTOG	
Lin [37]	2014	41	32—HR 9—very HR NCCN	42	45 Gy/25#	Yes 100%	92.7% 24 months	21 Gy/3# (prostate and SVs)	CK	91.9% (4 year bFFS)	≥Gr 3 GU 0% GI 0% Gr 2 GU 3–11% GI 0% CTCAE	Pilot study
Koh [38]	2014	8 in vHDRB cohort and 17 in SBRT monotherapy cohort	vHDRB cohort HR D’Amico	29.3 (entire cohort)	vHDRB cohort 40 Gy/20#	Unknown	vHDRB cohort 87.5% Unknown duration	18–24 Gy/3–5# (not reported)	CK	vHDRB cohort 50% bcPFS	vHDRB cohort ≥Gr 3 GU 0% GI 12.5% Gr 2 GU 0% GI 0% RTOG	Retrospective
			SBRT monotherapy cohort LR—HR D’Amico		SBRT monotherapy cohort 32–37.5 Gy/4–5#		SBRT monotherapy cohort 83.3% for IR 100% for HR Unknown duration			SBRT monotherapy cohort 100% bcPFS for LR-IR 83.3% bcPFS for HR	SBRT monotherapy cohort ≥Gr 3 GU 0% GI 0% Gr 2 GU 0% GI 0% RTOG	
Freeman [39]	2015	160 treated with boost among the total of 2000 in the study	819—LR 619—IR 172—HR 3—metastatic 130—unspecified NCCN	24	45–50 Gy/25#	Unknown	Unknown	19.5–21.75 Gy/3# (not reported)	CK	92%—entire cohort 87%—HR (2 year bcDFS)	≥Gr 3 GU 0% GI 0.1% Gr 2 not reported Unknown reporting scale	Prospective database, 92% of cohort were SBRT monotherapy patients
Mercado [40] and Paydar [41]	2016 and 2017	108	4—LR 45—IR 59—HR D’Amico	53	45–50.4 Gy/25–28#	No	63.6% 6 months	19.5 Gy/3# (prostate and proximal SVs)	CK	100%—IR 89.8%—HR (3 year bcPFS)	≥Gr 3 GU 6% GI 1% Gr 2 GU 40% GI 12% CTCAE	Retrospective
Pontoriero [42]	2016	5 in vHDRB cohort and 21 in SBRT monotherapy cohort	vHDRB cohort HR D’Amico	21.5 (entire cohort)	vHDRB cohort 46 Gy/23#	Yes (100%)	vHDRB cohort 100% 24 months	19 Gy/2# (not reported)	CK	vHDRB cohort 80% bcPFS	vHDRB cohort ≥Gr 2 GU 0% GI 0% CTCAE	Pilot study
			SBRT monotherapy cohort LR—IR D’Amico		SBRT monotherapy cohort 38 Gy/4#		SBRT monotherapy cohort 15% 6 months, 46% 24 months			SBRT monotherapy cohort 100% bcPFS	SBRT monotherapy cohort ≥Gr 2 GU 4.8% GI 4.8% CTCAE	
Kim [15,43,44,45]	2016 (Phak), 2017 (x2), and 2022	42	31—IR 11—HR NCCN	84.2	45 Gy/25#	Some	No	21 Gy/3# (prostate and SVs)	CK	100%—IR 77.8%—HR (8 year bFFS)	≥Gr 3 GU 0% GI 0% Gr 2 GU 11.9% GI 14.3% RTOG	Phase 1/2a
Pasquier/CKNO-PRO) [46,47]	2017 and 2020	76	IR D’Amico	62	46 Gy/23#	No	No	18 Gy/3# (prostate)	CK (N = 60) LINAC (N = 16)	87.4% (5 year bcRFS)	≥Gr 3 GU 0% GI 3.9% Gr 2 GU 1.4% GI 9.3% CTCAE	Phase 2, multicentre
Feng [48]	2018	145 in vHDRB cohort and 200 in SBRT monotherapy cohort	vHDRB cohort 5—LR 51—IR 89—HR D’Amico	vHDRB cohort 24	vHDRB cohort 45–50.4 Gy/25–28#	Unknown	vHDRB cohort 70.3% Unknown duration	19.5 Gy/3# (not reported)	CK	vHDRB cohort Not reported	vHDRB cohort 7.60 ± 0.42 AUA symptom score at 1 year 5.5% late urinary flare	Phase 1/2, 1 year AUA symptom scores significantly differed (*p* = 0.003)
			SBRT monotherapy cohort 75—LR 104—IR 21—HR D’Amico	SBRT monotherapy cohort 24	SBRT monotherapy cohort 35–36.25 Gy/5#		SBRT monotherapy cohort 12% Unknown duration			SBRT monotherapy cohort Not reported	SBRT monotherapy cohort 9.53 ± 0.47 AUA symptom score at 1 year 12% late urinary flare	
Alayed [49]	2019	30	IR NCCN	72	37.5 Gy/15#	No	3.3% <6 months	10/12.5/15 Gy in single # (prostate and SVs)	LINAC	92.3% bcPFS	≥Gr 3 GU 3.3% GI 3.3% ≥Gr 2 GU 43.3% GI 26.6% CTCAE	Phase 1 study
Eade/BOOSTER [50]	2019	36	13—IR 23—HR D’Amico	24	46 Gy/23#	Some (if HR)	61% 18 months	20/22/24 Gy in 2# (prostate) 25/27.5/30 Gy in 2# (to DIL if Identified)	LINAC	93.3% (3 year bFFS)	≥Gr 3 GU 0% GI 0% Gr 2 GU 19.3% GI 0% CTCAE	Phase 1 study
Johansson [51]	2019	504	94—LR 158—IR 135—HR 117—Very HR NCCN	113	50 Gy/25#	Some 16%—HR 60%—very HR	55% 17% LR 32% IR (5 months) 76% HR (9 months) 91% very HR (24 months)	20 Gy/4# (prostate and SVs)	Proton	100%, 94%—LR 94%, 87%—IR 82%, 63%—HR 72%, 55%—very HR (5 and 10 year PSA relapse-free)	≥Gr 3 GU 2% GI 0% (in pre-treatment symptom-free patients) Gr 2 not reported RTOG at 5 years	Proton boost, retrospective
Pryor/PROMETHEUS [52]	2019	135	103—IR 32—HR D’Amico	24	46 Gy/23# or 36 Gy/12#	Some (8%)	54% (36% <6 months and 18% >6 months)	19–20 Gy/2# (prostate and SVs)	LINAC	98.6% (2 year bcPFS)	≥Gr 3 GU 2.2% GI 2% Gr 2 GU 24.9% GI 4.5% CTCAE	Phase 2, multicentre
Pollack/LEAD [53]	2020	25	IR—HR NCCN	66	76 Gy/38#	Some (in HR, 56 Gy/38#)	56% 6 months	12–14 Gy/1# (MRI DIL)	Proton	92% bcPFS	≥Gr 3 GU 4% GI 0% Gr 2 GU 16% GI 16% CTCAE	Phase 1 using lattice extreme ablative dose technique
Wang [54]	2020	121 in vHDRB cohort and 132 in conventional cohort	HR—very HR NCCN	vHDRB cohort 48.5	vHDRB cohort 45 Gy/25# WPRT	Yes 100%	vHDRB cohort 91.7% Mean: 24.6 months	21 Gy/3# (prostate and SVs)	CK	vHDRB cohort 93.9% (4 year bFFS)	vHDRB cohort ≥Gr 3 GU 0.8% GI 1.7% Gr 2 GU 19.8% GI 1.7% CTCAE	Retrospective
				Conventional cohort 41.4	Conventional cohort 74–79.2 Gy in 1.8–2 Gy/#		Conventional cohort 97.7% Mean: 30.6 months			Conventional cohort 89.1% (4 year bFFS)	Conventional cohort ≥Gr 3 GU 2.3% GI 2.3% Gr 2 GU 15.9% GI 4.5% CTCAE	
Narang [55]	2020	44	11—HR 22—very HR 9—node-positive 2—metastatic NCCN	63.5	45 Gy to nodes, 50 Gy to prostate Boost involved nodes (54–56 Gy) All in 25#	Yes 100%	86.4% 3 months	18 Gy/3# (prostate) 16 Gy/2# (to bone metastatic lesions)	CK	91.4% (5 year bcPFS)	≥Gr 3 GU 4.5% GI 0% Gr 2 not reported CTCAE	Phase 1/2
Kim/ADEBAR [56]	2020	26	1—HR 25—very HR NCCN	35	44 Gy/20#	Yes 100%	100% 25 months	18—21 Gy/3# (prostate and SVs)	CK	88.1% (3 year bcRFS)	≥Gr 3 GU 0% GI 0% Gr 2 GU 4% GI 4% CTCAE	Phase 1/2
Milecki/HYPO-PROST [57]	2020	105 in vHDRB arm and 103 in conventional arm	HR D’Amico	60.1	vHDRB arm 46 Gy/23#	Yes	Yes 24 months	15 Gy/2# (prostate and SVs)	LINAC	vHDRB arm 78.2% (5 year bcRFS)	vHDRB arm ≥Gr 2 GU 5.9% GI 13.9% RTOG	Randomized, abstract only
					Conventional arm 76 Gy/38#					Conventional arm 82.9% (5 year bcRFS)	Conventional arm ≥Gr 2 GU 5.8% GI 8.6% RTOG	
Turna [58]	2021	34	HR D’Amico	41.2	50 Gy/25# WPRT	Yes 100%	88.2% 36 months	21 Gy/3# (prostate and proximal SVs)	LINAC	100% bcPFS	≥Gr 3 GU 0% GI 0% Gr 2 GU 8% GI 17.6% CTCAE	Retrospective
Chen [59]	2021	130 in vHDRB cohort and 101 in HDR brachytherapy boost cohort	38.2% IR 29% HR 32.8% very HR NCCN	vHDRB cohort 73.4	45 Gy/25#	Some	vHDRB cohort 96.2% 6 months	vHDRB cohort 19–21 Gy/2# (prostate and SVs)	CK	vHDRB cohort 88.8%, 85.3% (5 and 10 year BCRF)	vHDRB cohort ≥Gr 3 GU 4.6% GI 1.5% Gr 2 not reported RTOG	Propensity score-matched analysis with HDR brachytherapy boost
				HDR brachytherapy cohort 186			HDR brachytherapy cohort 92.1% 6 months	HDR brachytherapy cohort 19 Gy/2#		HDR brachytherapy cohort 91.8%, 74.6% (5 and 10 year BCRF)	HDR brachytherapy cohort ≥Gr 3 GU 3.0% GI 0% Gr 2 not reported RTOG	
Phuong [60]	2022	2 treated with vHDRB boost among the total of 22 in the study	IR—HR NCCN	32	41.25 Gy/15#	Yes 100%	95% 4 months	19 Gy/2# (vHDRB) 15 Gy/1# (HDR brachytherapy boost) (prostate and SVs)	CK	100% (3 year bcPFS)	≥Gr 3 GU 0% GI 0% Gr 2 GU 14% GI 5% CTCAE	Retrospective, mixed population with SBRT monotherapy
Novikov [61]	2022	51 in vHDRB cohort and 98 in HDR brachytherapy cohort	HR, very HR 29.4% of vHDRB cohort were node-positive 32.7% of HDR brachytherapy cohort were node-positive NCCN	vHDRB cohort 55.1	45–50.4 Gy/25–28# Some used 3dcrt	Yes 100%	100% Unknown duration	vHDRB cohort 21 Gy/3# (prostate and proximal SVs)	LINAC	vHDRB cohort 76.5%, 67.7% (3 and 5 year bcRFS)	vHDRB cohort ≥Gr 3 GU 0% GI 5.9% Gr 2 GU 9.8% GI 8.6% CTCAE	Retrospective, no rectal stabilization devices or hydrogel spacers used, significant Gr 3 and 4 toxicity in vHDRB cohort
				HDR brachytherapy cohort 87.7				HDR brachytherapy cohort 20 Gy/2# or 15 Gy/1#		HDR brachytherapy cohort 74.6%, 66.8% (3 and 5 year bcRFS)	HDR brachytherapy cohort ≥Gr 3 GU 1.1% GI 0% Gr 2 GU 28.6% GI 8.2% CTCAE	

**Abbreviations**: N, number of patients; ADT, androgen deprivation therapy; WPRT, whole pelvic radiotherapy; SV, seminal vesicle; CK, CyberKnife; LINAC, linear acceleration; LR, low risk; IR, intermediate risk; HR, high risk; GI, gastrointestinal; GU, genitourinary; SBRT, sterotactic body radiation therapy; HDR, high-dose-rate brachytherapy; DIL, dominant intraprostatic lesion; AUA, American Urological Association; bFFS, biochemical failure-free survival; bcPFS, biochemical progression-free survival; bcRFS, biochemical recurrence-free survival; bcDFS, biochemical disease-free survival; bNED, biochemical no evidence of disease; BCRF, biochemical recurrence freedom; RTOG/CTCAE indicate that the RTOG or Common Terminology Criteria for Adverse Events toxicity scales were used for reporting.

**Table 3 cancers-15-02018-t003:** Future stereotactic boost trials identified from the ClinicalTrials.gov database.

Trial Number	Phase	Location	Dose	Accrual Number
NINJA/TROG 18.01	Phase 3 (randomized)	Trans-Tasman Radiation Oncology Group (TROG)	monoSBRT of 40 Gy/5# vs. conventional vHDRB of 36 Gy/12# + boost of 20 Gy/2#	472
NCT01618851	Phase 2	Georgetown University	Conventional treatment of 45 Gy/25# + boost of 19.5 Gy/3#	70
NCT03564275	Prospective/matched-pair analysis	University of Cincinnati	Proton boost, no details on dosing	50
NCT01508390	Phase 2	Boston Medical Center	Standard radiotherapy + CK boost of 21 Gy/3#	35
NCT03380806	Phase 2 (randomized)	Juravinski Cancer Center	45 Gy/25# conventional treatment then randomized to either 33–35 Gy/16# or SBRT boost of 19.5–21 Gy/3#	100
NCT02016248	Prospective	MemorialCare Health System	50.4 Gy/28# then CK boost of 27.5 Gy/5#	167
NCT01839994	Phase 3	Maria Sklodowska-Curie National Research Institute of Oncology	Experimental arm: conventional 50 Gy/25# then boost (brachytherapy or SBRT of 20 Gy/2#)	350
NCT02672449	Phase 2	European Institute of Oncology	Conventional 45 Gy/25# + carbon ion boost (16.6 GyE/4#)	65
NCT03778112	Prospectiverandomized	Rush University Medical Center	SBRT to whole prostate (36.25 Gy/5#) vs. 45 Gy/25# conventional treatment + 18–21 Gy/3# boost	58
NCT01985828	Prospective	Advocate Health Care	45–50.4 Gy in 25–28# + CK boost of 21 Gy/3#	72
NCT02307058	Phase 2 (randomized)	University of Miami	76 Gy/38# + LEAD (proton boost) of 12–14 Gy/1# vs. 91.2 Gy/39#	164
NCT02064036	Not stated	University of California	No details on dose	29
NCT02339948	Phase 2	Genesiscare USA	monoSBRT of 40 Gy/5# vs. 45 Gy/25# conventional treatment + boost of 22 Gy/4#	279
NCT01352598	Prospective	Mercy Research	Conventional treatment (dose not specified) + 19–21 Gy/3# boost	84

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
