# Peer review of "Virtual HDR Boost for Prostate Cancer: Rebooting a Classic Treatment Using Modern Tech"

_cancers, 2023, doi:10.3390/cancers15072018_

Round 1

Reviewer 1 Report

In this review authors compared the results between the classical brachytherapy and the virtual high dose brachytherapy (vHDRB) as a propedeutic treatment for radiotherpay in prostate cancer basing on the published literature. Authors selected 34 studies conducted in many countries by many other authors using as a selection criteria the presence of the keywords “SABR AND prostate” OR “Stereotactic AND prostate” OR “SBRT AND prostate” OR “Boost AND prostate” in article title or abstract and as exclusion criteria the use of stereotactic monotherapy, simultaneous integrated boost (SIB) or radiotherapy delivered in the salvage setting. Authors analyzed in particular the diffences between the two techniques regarding the biochemical progression free survival and the toxicities, also analyzing the differences in the imaging and the radiation dose between the two treatments.

My suggestions:

Toxicities are reported with different scores so the evidence of the difference of toxicity is not very strong. Please standardize with a single score the entity of the toxicity

Not all analyzed aspects are considered basing on the risk of the PC. Please consider it for every outcome to standardize the results

Tumor stadiation in the selected studies is operated with TC and bone scan. Updated staging techniques would give a stronger evidence to the work. At this regard i can suggest you the analisys of this work https://pubmed.ncbi.nlm.nih.gov/35814914/

In paragraph 5.1 is said that “There is no strong randomized data to compare a HDR brachytherapy boost to vHDRB”. RCT are required to improve the evidence resulting from this review

Understanding the actual anatomy of the prostate is a crucial node to correctly diagnose PC, to stratify cancer risk and to follow it up. At this regad i can suggest the analisys of this work https://pubmed.ncbi.nlm.nih.gov/34247169/

Author Response

Thank you, Expert Reviewer, for your time, expertise and effort into improving this manuscript. Your valuable contributions and improvements have been duly noted. In response to your comments ...

My suggestions:

Toxicities are reported with different scores so the evidence of the difference of toxicity is not very strong. Please standardize with a single score the entity of the toxicity

You raise an important and valid point, and sadly a problem amongst radiotherapy trials and studies. The different scoring systems (mainly CTCAE and RTOG) are not interchangeable and as such it was important to report these different grading scale used. Currently there is no proposed standardised toxicity reporting scale and we believe this to be an important issue but one that is outside the scope of this manuscript. We did try to account for these caveats in the last paragraph of section 5.2 and in particular

“Toxicity data from all these studies have multiple contributing factors not easily teased out. An example is pelvic nodal irradiation and the effect on GI toxicity. Some combine G2+ toxicity whereas others divide G2 and G3. There is a difference in scoring and statistical calculation methods. Some series report toxicity as a prevalence at a specific time-point versus others citing a cumulative toxicity result. Overall, this data supports the safety profile for using vHDRB for PC, and may have lower late toxicities compared to brachytherapy. The main caveat is much of the data comes from relatively small, single institution series, and is non-randomized.”

Not all analyzed aspects are considered basing on the risk of the PC. Please consider it for every outcome to standardize the results

Thank you and again an important / vital point you raise. We do agree about this important point and have put in future directions the overarching statement of

“High quality prospective randomized data is needed before vHDRB is considered standard of care.”

Sadly most of these studies did not segment out their PC risk, and even amongst this some used D’Amico staging and others NCCN. To try and at least give insight into this difference across risk groups, we were able to tease out of each article the numbers of HR, IR and LR  and  if the studies included differences of outcomes based on risk we did include them in Table 2. However due to different staging used, different techniques and doses and accounting for low patient numbers there lacks an ability to standardize the results. Even the most basic pathology grading has evolved during this time making no true ability to standardize (https://www.ncbi.nlm.nih.gov/pmc/articles/PMC6375094/).

Tumor stadiation in the selected studies is operated with TC and bone scan. Updated staging techniques would give a stronger evidence to the work. At this regard i can suggest you the analisys of this work https://pubmed.ncbi.nlm.nih.gov/35814914/

Thank you for this valuable contribution and we have referenced this contribution and important point.

“Most of these studies staged patients with conventional CT and bone scan. Radiomics has the potential to offset some of the poor sensitivity and specificity issues [62].” PSMA PET staging has been shown to be superior compared to conventional staging [63] and future trials may be expected to have better biochemical control due to improved patient selection. “

In paragraph 5.1 is said that “There is no strong randomized data to compare a HDR brachytherapy boost to vHDRB”. RCT are required to improve the evidence resulting from this review

Thank you have added to the sentence “and future randomized control trials are needed to answer this”.

This is discussed also further in future directions (section 8) when discussing the trial NCT01839994 which will hopefully answer this question.

Understanding the actual anatomy of the prostate is a crucial node to correctly diagnose PC, to stratify cancer risk and to follow it up. At this regad i can suggest the analisys of this work https://pubmed.ncbi.nlm.nih.gov/34247169/

Thank you for this, this article seemed to have escaped my sight, however asks some intriguing questions. Would agree that radiomics is next in the evolution of prostate cancer staging and if image pre-processing is standardized deep learning AI could be implemented to help revolutionize this poor prognostication stratification dilemma we face in PC.  Unfortunately the radiomics aspect on prostate volume is a bit outside the scope of this clinical review article.

Thank you again for your valuable contributions, time and skill reviewing this manuscript. 

Reviewer 2 Report

The study aimed to consider the evidence behind a brachytherapy boost for PC, the continuing evolution of vHDRB approaches, culminating in the current generation of clinical trials which will help define the role of this emerging modality. Following are some suggestions to improve your manuscripts.

- The authors should add information related to “Author, Contributions, and Funding, Conflicts of Interest”.

- Introduction: The authors should rewrite these sentences to make clearer:

Higher radiotherapy dosage correlate with greater tumor control probability (figure 1), with several randomized trials showing improved disease control outcomes with higher delivered radiotherapy doses [2]”.

In this review we focus on the biology and clinical evidence behind dose escalation for PC, the evidence but conversely declining use of brachytherapy to achieve this, and the investigations into SBRT as an alternative to brachytherapy termed virtual high dose rate brachytherapy boost (vHDRB).

“Studies were also excluded if significant details were missing, for example [11] a multi-institutional patient registry which had 437 patients, however only 5% of these patients received a stereotactic boost and no further conclusions could be drawn on this patient population.” -> The authors should rewrite: “….; however, ….

- In section “2. Radiation Dose escalation – theory and evidence”:

ü  The authors should rewrite this sentence to make clearer: “To understand this, understanding of the concept of Biologically Effective Dose (BED) is useful”.

ü  There is a redundant word “that” in this sentence: “The main caveat is that that much of the data comes from relatively small, single institution series, and is non-randomized”. Please remove it.

- The authors should remove redundant dots after sections 5, 6, and 8.

- In section 5.2: “7.60 ± 0.42 and 9.53 ± 0.47, p = 0.003” should be enclosed in bracket.

- After reference [70], there is an unnecessary highlight. Please double-check and remove it.

- In section 7.3:

ü  This sentence should be rewritten: “This trial showed with hydrogel insertion rectal dose was decreased and there was a significant reduction in late rectal toxicity, 2.0% compared to 7.0% in the control group (p = 0.04)”

ü  Please convert Italic font in the below paragraph into normal: “Endorectal devices were utilized in some of these trials to help minimize rectal toxicity ... It was also seen that one-third of patients the rectal balloon had a systematic error due to air leakage, suboptimal inflation or suboptimal rectal emptying.

- In section 7.4: “The PACE-B trial [10] mentions some differences in toxicities seen between CyberKnife and standard linear accelerator toxicity, however the low numbers precluded statistical analysis” -> This sentence should be revised: the PACE-B trial [10] mentions some differences in toxicities seen between CyberKnife and standard linear accelerator toxicity; however, the low numbers precluded statistical analysis.

- In section 8. This sentence should be rewritten to be clearer: Four randomized trials stand out as helping evaluate the true impact of a vHDRB -> Four randomized trials stand out as helping evaluate the true impact of a vHDRB on PC???

- It would be better if another English editor can make it smoother.

Author Response

Thank you, Expert Reviewer, for your time, expertise and effort into improving this manuscript. Your valuable contributions and improvements have been duly noted. 

In response to your suggestions: 

- The authors should add information related to “Author, Contributions, and Funding, Conflicts of Interest”.

-Thank you have added

Introduction: The authors should rewrite these sentences to make clearer:

“Higher radiotherapy dosage correlate with greater tumor control probability (figure 1), with several randomized trials showing improved disease control outcomes with higher delivered radiotherapy doses [2]”.

-Thank you have rephrased this.

“Increased radiotherapy doses have been found to be associated with higher tumour control probability (figure 1). This relationship has been demonstrated by several randomized trials, which have reported better disease control outcomes with higher radiotherapy doses”

“In this review we focus on the biology and clinical evidence behind dose escalation for PC, the evidence but conversely declining use of brachytherapy to achieve this, and the investigations into SBRT as an alternative to brachytherapy termed virtual high dose rate brachytherapy boost (vHDRB).

-Thank you. We have rephrased this.

“In this review we focus on the biology and clinical evidence behind dose escalation for PC aiming at a target audience of all clinicians involved in the management of this disease. Furthermore, we explore the evidence but conversely declining use of brachytherapy to achieve this, and the investigations into SBRT as an alternative to brachytherapy termed virtual high dose rate brachytherapy boost (vHDRB).“

“Studies were also excluded if significant details were missing, for example [11] a multi-institutional patient registry which had 437 patients, however only 5% of these patients received a stereotactic boost and no further conclusions could be drawn on this patient population.” -> The authors should rewrite: “….; however, ….

-Thank you have rewritten for clarity “Studies that lacked significant details were also excluded from the analysis. For instance, in a multi-institutional patient registry study [11] that involved 437 patients, only a small proportion (5%) of the patients received a stereotactic boost, and therefore, no conclusive findings could be drawn regarding this subgroup.”

- In section “2. Radiation Dose escalation – theory and evidence”:

The authors should rewrite this sentence to make clearer: “To understand this, understanding of the concept of Biologically Effective Dose (BED) is useful”.

-Thank you have rewritten “To comprehend this concept, it is helpful to have an understanding of the Biologically Effective Dose (BED).”

There is a redundant word “that” in this sentence: “The main caveat is that that much of the data comes from relatively small, single institution series, and is non-randomized”. Please remove it.

-Thank you have removed

The authors should remove redundant dots after sections 5, 6, and 8.

-Thank you have corrected, the original formatting seems to have changed. Thank you again for your insights.

In section 5.2: “7.60 ± 0.42 and 9.53 ± 0.47, p = 0.003” should be enclosed in bracket.

-Thank you have corrected.

After reference [70], there is an unnecessary highlight. Please double-check and remove it.

-Thank you and have corrected. Despite many orders of eyes reviewing, no one picked this. Thank you very much for your attention to detail.

In section 7.3:

This sentence should be rewritten: “This trial showed with hydrogel insertion rectal dose was decreased and there was a significant reduction in late rectal toxicity, 2.0% compared to 7.0% in the control group (p = 0.04)”

-Thank you have clarified to “The trial demonstrated a decrease in rectal dose and a significant reduction in late rectal toxicity (2.0% versus 7.0% in the control group, p=0.04) with the use of hydrogel insertion.”

-Please convert Italic font in the below paragraph into normal: “Endorectal devices were utilized in some of these trials to help minimize rectal toxicity ... It was also seen that one-third of patients the rectal balloon had a systematic error due to air leakage, suboptimal inflation or suboptimal rectal emptying.

Thank you, have corrected.

In section 7.4: “The PACE-B trial [10] mentions some differences in toxicities seen between CyberKnife and standard linear accelerator toxicity, however the low numbers precluded statistical analysis” -> This sentence should be revised: the PACE-B trial [10] mentions some differences in toxicities seen between CyberKnife and standard linear accelerator toxicity; however, the low numbers precluded statistical analysis.

-Thank you, have corrected.

In section 8. This sentence should be rewritten to be clearer: Four randomized trials stand out as helping evaluate the true impact of a vHDRB -> Four randomized trials stand out as helping evaluate the true impact of a vHDRB on PC???

-Thank you, have corrected per your suggestion.

It would be better if another English editor can make it smoother.

-Thank you for your corrections and excellent contributions. We went through the manuscript again with two first language English authors and appreciate your help improving the clarity of the manuscript.

Reviewer 3 Report

General comments:

The review on virtual HDR boost for prostate cancer is of interest and quite comprehensive. While generally well-written, the paper needs a revision for a better focus of the topic.

(1)   There is a lot of basic information throughout the paper that could be removed or written more succinctly (such as the information on therapeutic ratio, BED, description of Cyberknife, etc).

(2)   Introduction – add a paragraph explaining boost brachytherapy vs monotherapy. Explain why your selection criteria for studies includes boost-only. Comment on the current role of LDR/HDR as monotherapy.

(3)   I don’t see the role of table 1 on BED comparison across various fractionation schedules, as BED is not discussed further in the review.

Specific comments:

1.     Title – replace ‘tech’ with ‘techniques’.

2.     Figure 1 is basic knowledge and adds no value to the paper – please remove.

3.     Introduction – “fractions” does not need quotation marks – it is a well acknowledged terminology for fractionated radiotherapy, whether EBRT or brachytherapy

4.     Rename ‘box 1’ as table 1 or figure 1

5.     Section 3 – replace ‘comparted’ with ‘compared’

6.     Section 5.1 – rephrase / join the statements into a stand-alone phrase: ‘Certain questions and challenges remain in defining outcomes and measurable endpoints: first, the varied dose schedules and dose distributions, and second, cofounders such as ADT use / duration and pelvic nodal radiotherapy inclusion.

7.     Section 5.2 – last paragraph – replace ’verses’ with ‘versus’

8.     Section 5.2 – last paragraph – remove ‘The main caveat is that much of …’ (‘that’ is duplicated)

9.     Section 6 – this statement doesn’t have a verb “A purposeful aim to create ‘hot-spots’ within defined volumes receiving up to 150% of the prescribed dose.” – either rephrase or join it with the previous sentence.

10.  Section 7.3. should not be italic

11.  Section 7.3. – ‘It was also seen that in/for one-third of patients the rectal balloon..’

Author Response

Thank you, Expert Reviewer, immensely for reviewing this manuscript. Your time, expertise and effort into improving this manuscript is vastly appreciated and we appreciate your valuable contributions.

The deep insights into this manuscript are evident in the questions/comments you discuss and interestingly were the same points debated amongst the group early on in creation.    

The review on virtual HDR boost for prostate cancer is of interest and quite comprehensive. While generally well-written, the paper needs a revision for a better focus of the topic.

(1)   There is a lot of basic information throughout the paper that could be removed or written more succinctly (such as the information on therapeutic ratio, BED, description of Cyberknife, etc).

-We think the main discrepancy on this manuscript stems from the lack of our description of target audience. We would agree that there is a lot of basic information that could be omitted if the audience is an expert / radiation oncologist specializing in prostate radiotherapy. Which we agree, the majority of readers will be. However, in doing so we may alienate some important readers. These would be urologists, medical oncologists and early radiation oncologists/registrars who have limited understanding behind the theory of HDR boosting and the technologies behind this. This is the main reason behind one of our authors being a lead local urologist. The creation of vHDRB was founded in strong radiobiology principles and to negate this we think the readers would miss out on the most important part, the rationale.

To clarify this, we have changed the introductory statement to better reflect our intended target audience.

“In this review we focus on the biology and clinical evidence behind dose escalation for PC aiming at a target audience of all clinicians involved in the management of this disease. Furthermore, we explore the evidence but conversely declining use of brachytherapy to achieve this, and the investigations into SBRT as an alternative to brachytherapy termed virtual high dose rate brachytherapy boost (vHDRB). “  

(2)   Introduction – add a paragraph explaining boost brachytherapy vs monotherapy. Explain why your selection criteria for studies includes boost-only. Comment on the current role of LDR/HDR as monotherapy.

 -added into 1. Introduction -paragraph 2

“Brachytherapy monotherapy is a proven treatment modality for favourable intermediate risk PC, based on NCCN treatment guidelines, with excellent oncological outcomes [3,4]. Higher risk PC necessitates a combined modality approach with brachytherapy boost and EBRT due to risk of disease outside the prostatic capsule.“

-added in 1.1 – selection criteria

“Only boost studies were selected for review as this radiotherapy technique has emerged more recently. Both these techniques developed alongside one another, yet we await the results of randomized trials to adequately compare these two [13]”

   -added in section 3

Around a century ago, brachytherapy using radium, which involved permanent low dose rate (LDR) implants, emerged as the first successful radiotherapy modality for prostate cancer treatment. Despite the changes in isotopes used, LDR monotherapy has consistently demonstrated excellent oncological outcomes in patients with favorable intermediate risk PC. In the 1980s, as external beam radiation therapy (EBRT) techniques such as 3D conformal radiotherapy evolved, brachytherapy also saw technological advancements, including the introduction of temporary high dose rate (HDR) insertions. Following these developments, the combination of these two techniques was pursued to improve oncological outcomes in patients with higher risk PC. The approach involves using brachytherapy to intensify dose in areas with higher disease burden, while delivering lower doses of EBRT to microscopic areas. This approach has shown great promise in improving oncological outcomes in patients with higher risk PC.

(3)   I don’t see the role of table 1 on BED comparison across various fractionation schedules, as BED is not discussed further in the review.

- We agree an expanded discussion on BED is important to provide appropriate context for table 1 and have expanded the paragraph to describe this more fully.

“However, in radiotherapy not every Gray (Gy) may be equal, and new radiotherapy methods were trialed to deliver dose escalation through different approaches. To comprehend this concept, it is helpful to have an understanding of the Biologically Effective Dose (BED). In short, a higher dose per fraction size can deliver a higher BED compared to conventional EBRT dose schedules of 2 Gy per fraction. Each tissue and tumour has its own radiation fraction size sensitivity, described by the alpha-beta ratio. By using a higher dose per fraction size, theoretically this should result in a higher biological dose to the tumour and lead to greater tumour cell death.  See table 1 for an example of BED dosing for different delivered fraction sizes.   “

Specific comments:

  1. Title – replace ‘tech’ with ‘techniques’.

-This word was meant to be polysemy with meanings of technology and techniques (both which relate to the creation of vHDR) and for this reason we would respectfully ask to remain with our original title.

  1. Figure 1 is basic knowledge and adds no value to the paper – please remove.

  -As discussed in major comment (1) above and if viewed in context of target audience, this general knowledge figure we think would add value to the description.

  1. Introduction – “fractions” does not need quotation marks – it is a well acknowledged terminology for fractionated radiotherapy, whether EBRT or brachytherapy

-Thank you, have removed

  1. Rename ‘box 1’ as table 1 or figure 1

-Thank you, have corrected

  1. Section 3 – replace ‘comparted’ with ‘compared’

-Thank you, have corrected

  1. Section 5.1 – rephrase / join the statements into a stand-alone phrase: ‘Certain questions and challenges remain in defining outcomes and measurable endpoints: first, the varied dose schedules and dose distributions, and second, cofounders such as ADT use / duration and pelvic nodal radiotherapy inclusion.

-Thank you, have corrected and appreciate the flow improvements.

  1. Section 5.2 – last paragraph – replace ’verses’ with ‘versus’

-Thank you, have corrected and again appreciate your eyes to these details.  

  1. Section 5.2 – last paragraph – remove ‘The main caveat is that much of …’ (‘that’ is duplicated)

-Thank you, have corrected

  1. Section 6 – this statement doesn’t have a verb “A purposeful aim to create ‘hot-spots’ within defined volumes receiving up to 150% of the prescribed dose.” – either rephrase or join it with the previous sentence.

Thank you, have corrected and joined the prior sentence.

  1. Section 7.3. should not be italic

Thank you, sorry, I think the formatting changed somewhere, have corrected.

  1. Section 7.3. – ‘It was also seen that in/for one-third of patients the rectal balloon..’

-Thank you, have corrected

Thank you again Expert Reviewer, your time, skill and knowledge has helped improve this manuscript significantly. We are very thankful for your contributions. 

Round 2

Reviewer 1 Report

Authors answered all comments and suggestions.

Reviewer 3 Report

The authors have satisfactorily addressed all comments raised by this reviewer.